# Breaking Modality Gap in RGBT Tracking: Coupled Knowledge Distillation

### Andong Lu
Information Materials and Intelligent Sensing Laboratory of Anhui Province, Anhui Provincial Key Laboratory of Multimodal Cognitive Computation, School of Computer Science and Technology, Anhui University.
Hefei, Anhui, China
adlu_ah@foxmail.com

### Jiacong Zhao
Information Materials and Intelligent Sensing Laboratory of Anhui Province, Anhui Provincial Key Laboratory of Multimodal Cognitive Computation, School of Artificial Intelligence, Anhui University.
Hefei, Anhui, China
jiacongzhao2022@163.com

### Chenglong Li*
Information Materials and Intelligent Sensing Laboratory of Anhui Province, Anhui Provincial Key Laboratory of Security Artificial Intelligence, School of Artificial Intelligence, Anhui University.
Hefei, Anhui, China
lcl1314@foxmail.com

### Yun Xiao
Anhui Provincial Key Laboratory of Multimodal Cognitive Computation, School of Artificial Intelligence, Anhui University.
Hefei, Anhui, China
xiaoyun@ahu.edu.cn

### Bin Luo
Anhui Provincial Key Laboratory of Multimodal Cognitive Computation, School of Computer Science and Technology, Anhui University.
Hefei, Anhui, China
luobin@ahu.edu.cn

## Abstract

Modality gap between RGB and thermal infrared (TIR) images is a crucial issue but often overlooked in existing RGBT tracking methods. It can be observed that modality gap mainly lies in the image style difference. In this work, we propose a novel Coupled Knowledge Distillation framework called CKD, which pursues common styles of different modalities to break modality gap, for high performance RGBT tracking. In particular, we introduce two student networks and employ the style distillation loss to make their style features consistent as much as possible. Through alleviating the style difference of two student networks, we can break modality gap of different modalities well. However, the distillation of style features might harm to the content representations of two modalities in student networks. To handle this issue, we take original RGB and TIR networks as the teachers, and distill their content knowledge into two student networks respectively by the style-content orthogonal feature decoupling scheme. We couple the above two distillation processes in an online optimization framework to form new feature representations of RGB and thermal modalities without modality gap. In addition, we design a masked modeling strategy and a multi-modal candidate token elimination strategy into CKD

to improve tracking robustness and efficiency respectively. Extensive experiments on five standard RGBT tracking datasets validate the effectiveness of the proposed method against state-of-the-art methods while achieving the fastest tracking speed of 96.4 FPS.

## CCS Concepts

• **Computing methodologies** → **Tracking**.

## Keywords

RGBT Tracking, cross-modality distillation, modality gap

**ACM Reference Format:**

Andong Lu, Jiacong Zhao, Chenglong Li, Yun Xiao, and Bin Luo. 2024. Breaking Modality Gap in RGBT Tracking: Coupled Knowledge Distillation. In *Proceedings of the 32nd ACM International Conference on Multimedia (MM '24), October 28-November 1, 2024, Melbourne, VIC, Australia.* ACM, New York, NY, USA, 10 pages. https://doi.org/10.1145/3664647.3680878

## 1 Introduction

In recent years, the field of RGBT tracking attracts great attention due to its wide range of applications in surveillance, object recognition and other fields [1, 6, 17, 27, 28, 40, 42, 53, 56, 57]. RGBT tracking aims to take advantage of the complementary advantages of RGB and thermal modalities to achieve robust object tracking. However, visible spectrum and thermal infrared data are collected by cameras in different imaging bands, reflecting different properties of the target object. As a result, they differ significantly in appearance style, which inevitably leads to the issue of modality gap.

Current research on RGBT tracking focuses on three categories, including multi-modal fusion design, multi-modal representation learning, and prompt learning. The first category of studies [26, 27] are usually devoted to designing a reliable late fusion module for

---

*Chenglong Li is the corresponding author.

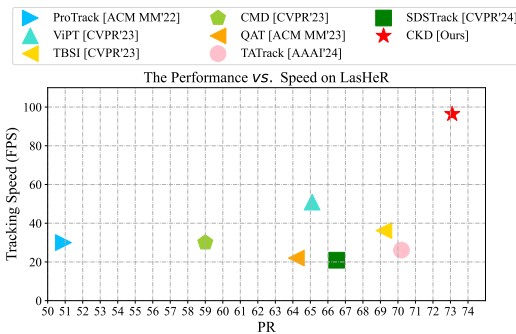

Figure 1: Comparison of performance and speed for state-of-the-art tracking methods on LasHeR [23]. We visualize the Precision Rate (PR) to the Frames Per Second (FPS). CKD is able to rank the 1st in PR while running at 96.4 FPS.

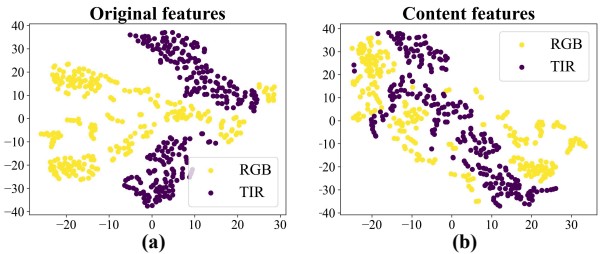

Figure 2: Illustration of the influence of modality style on modality gap. Here, (a) denotes the feature distribution of the two modalities, and (b) denotes the feature distribution of the two modalities after removing the style information using instance normalization.

the collaboration between RGB and thermal infrared (TIR) modality features. For instance, Liu *et al.* [26] proposes a quality-aware fusion module that carefully designs two independently weighted prediction branches to guide the fusion of multi-modal features. The second category of research [5, 17, 28, 45] focus on integrating multi-layer feature interaction modules into backbone network, thus can leverage modality complementary information to enhance the representation of each modality. For example, Hui *et al.* [17] introduce a Transformer-based multi-layer feature interaction module, which effectively enhances the exchange of information between modalities. The third category of methods [1, 47, 58] explore the concept of prompt learning in multi-modal feature interaction. For example, Cao *et al.* [1] design a bi-directional adapter for mutual prompting of modality information. However, existing studies have long overlooked the influence of modality gap, thus limiting the performance and efficiency of current RGBT tracker.

To address this challenge, we propose a novel Coupled Knowledge Distillation framework (CKD), which pursues common styles of different modalities to break modality gap, for high performance RGBT tracking. Figure 1 presents the advantages of CKD compared to existing state-of-the-art methods in two different dimensional metrics, which suggest a powerful potential in breaking modality gap. Specifically, we analyse the impact of modality style on the modality gap in Figure 2, which can be seen that the modality gap is significantly reduced after removing the modality style. Therefore, the modality style plays an important factor in breaking modality gap. However, the structure of the modal feature distribution is also affected, which may harm the modal content representation.

To this end, we introduce two student networks and design a style distillation scheme between the style features of the two students to make their style features as consistent as possible, aiming to eliminate the modality gap. Here, we utilize the feature mean and standard deviation to represent the style features of a modality, which has been proven to be effective in many studies [15, 16]. However, style feature distillation may harm modality content representations of both student branches. We further introduce the original RGB and TIR networks as teachers, and design a content distillation scheme to pursue the consistency between the content features of the teacher branch and the corresponding student

branch. To alleviate the constraints on student style features, we employ a style-content orthogonal feature decoupling strategy that obtains content features by performing instance normalization on the original features. Consequently, we couple the above two distillation processes in an online optimization framework to pursue the common style of the two modalities while avoiding the damage to the modality content representation.

In addition, we design a masked modeling strategy to further enhance the robustness of modality features in challenging scenarios. It involves the random mask to create data pairs of content-degraded and non-degraded of one modality, and then feed into the teacher and student branches. We can employ content distillation loss in CKD to effectively learn representations for content reconstruction. We also design a simple and effective multi-modal candidate token elimination strategy, which collaboratively considers the information of the two modalities to jointly decide the candidate elimination tokens in the search region. By drop these tokens during the inference phase, we can achieve a balance between tracking performance and tracking efficiency.

In summary, our major contributions are as follows.

- We propose a novel coupled knowledge distillation framework CKD to handle the modality gap issue by eliminating the style difference between RGB and thermal images for high performance RGBT tracking. To the best of our knowledge, this research is the first effort to break the modality gap in RGBT tracking.
- We design a style-content coupled distillation scheme based on style-content orthogonal feature decoupling, which effectively eliminates modality gap without harming the modality content representation.
- We present a masked modeling scheme that is seamlessly integrated into CKD, which effectively enhances the learning of modality content representation. In addition, a multi-modal candidate token elimination strategy is designed, which further improves the tracking efficiency.
- The proposed method achieves an impressive tracking speed of 96.4 FPS while achieving state-of-the-art results on four mainstream public datasets. Compared with the existing methods, the PR/SR scores on RGBT210, RGBT234, LasHeR and VTUAV datasets are increased by 1.6%/2.7%, 1.6%/3.0%,

3.0%/2.0% and 10.1%/11.1%, respectively, and the speed is increased by 60.2 FPS.

## 2 Related work

### 2.1 RGBT Tracking

RGBT tracking, as a rapidly growing research field, has witnessed a significant surge in the development of innovative algorithms. Existing studies can be broadly classified into three categories, including multi-modal fusion design, multi-modal representation learning and prompt learning. The first category [26, 37, 50, 62] is dedicated to designing diverse fusion strategies that combine the features of two modalities. Liu *et al.* [26] employ feature weighted fusion architectures based on the importance or quality of each modality. Tang *et al.* [37] apply modal weighting fusion strategy to three different levels of pixel, feature and decision. However, the optimal level of fusion varies in different scenarios. The second category [17, 22, 27, 28, 45, 51] is devoted to the improvement of modal representation by feature decoupling or feature interaction. For instance, Li *et al.* [22, 27] design a multi-adapter framework to simultaneously extract modality-shared and modality-specific features. Similarly, Zhang *et al.* [51] design a multi-branch challenge framework to model modality representation under different challenge scenarios. Lu *et al.* [28] design multi-layer feature interaction modules to enhance the modality representation. The third category [1, 12, 14, 47, 58] focuses on modal information interaction in a prompting manner, and is also a recent hot research topic. For example, Yang *et al.* [47] and Zhu *et al.* [58] introduce the concept of prompt learning at the pixel level and the feature level of RGBT tracking, respectively, for efficient multi-modal tracking. However, these methods ignore the impact of modality gap, which limits the RGBT tracking performance.

### 2.2 Knowledge Distillation

Knowledge distillation [11] aims to transfer the knowledge from a pre-trained large-scale teacher model to a small-scale student model. According to the type of knowledge transferred, the current research is mainly divided into three categories. Probability-based methods [11, 36] leverage the teacher model's predicted probabilities, guiding the student model to emulate the teacher's log-class distribution through minimizing KL divergence. Conversely, Feature-based methods [3, 9, 49] harness the mid-layer output feature map of the network for supervising the training of the student model. Relation-based methods [32, 59] emphasize inter-sample relationships over single instances. While simple to implement, it requires complex teacher modeling and significant training time. To address these issues, some studies [34, 35] utilize a mutual learning strategy for knowledge transfer. In addition, cross-modality knowledge distillation aims to transfer knowledge between different modalities. Some studies [33, 54] use dominant modality to guide weaker ones, while others [7, 18] investigate inter-modal complementarity. In contrast to existing methods, the coupled knowledge distillation method proposed in this paper distills coupled modality content and modality style knowledge from different modalities, thus facilitating modality style transfer for modality content preservation achievable by student models.

## 3 Methodology

In this section, we first introduce the overall architecture of Coupled Knowledge Distillation (CKD) and then describe the coupled knowledge distillation approach. Subsequently, mask modeling and multi-modal candidate token elimination are introduced. Finally, the training process and implementation details are given.

### 3.1 Framework Overview

We provide a detailed description of CKD framework, and its overall structure is shown in Figure 3. During the training phase, our CKD framework comprises four branches (including a RGB teacher branch, a TIR teacher branch, a RGB student branch, and a TIR student branch) as well as three tracking heads (including a multi-modal tracking head and two single-modal tracking heads). In the testing phase, CKD consists of two student branches and one multi-modal tracking head. In specific, for the given RGB and TIR modal frames, the search and template frames are first partitioned into patches with the size of $p \times p$ using four independent learnable patch embedding layers, and flattened to obtain four search token sequences $(S_{rgb}^t, S_{tir}^t, S_{rgb}^s, S_{tir}^s)$, and four template token sequences $(T_{rgb}^t, T_{tir}^t, T_{rgb}^s, T_{tir}^s)$. Following [48], we also add learnable position embeddings to the above tokens to provide positional prior information. Note that we add random mask to the search frame token sequences $(S_{rgb}^s, S_{tir}^s)$ in the two student branches during training.

Then, we concatenate search and template frame token sequences for the four groups, denoted as $I_{rgb}^t = [S_{rgb}^t, T_{rgb}^t]$, $I_{tir}^t = [S_{tir}^t, T_{tir}^t]$, $I_{rgb}^s = [S_{rgb}^s, T_{rgb}^s]$, and $I_{tir}^s = [S_{tir}^s, T_{tir}^s]$. We feed $I_{rgb}^t$ and $I_{tir}^t$ into the RGB and TIR teacher branches, respectively, while $I_{rgb}^s$ and $I_{tir}^s$ into the RGB and TIR student branches, respectively. These branches share an identical structure, consisting of standard Transformer blocks [8], but they have independent parameters. Finally, we concatenate the last layer features of the RGB and TIR student branches in the channel dimension and input them into the tracking head [48] for object localization and regression. The last layer features of the RGB and TIR teacher branches are separately fed into two independent tracking head [48] networks for task learning.

### 3.2 Coupled Knowledge Distillation

The proposed CKD is the first effort in the field of RGBT tracking to address modality gap. In particular, CKD employs a style distillation to make the style features of the two modalities as consistent as possible, thus breaking modality gap. It also introduces a content distillation to ensure that the modality content representation is stable. Next, we describe two distillation methods, namely style distillation and content distillation, in detail.

*3.2.1 Style Distillation.* Style distillation aims at mutual distillation between the style features of the two student branches, thus pursuing a common style for both modalities to break modality gap. Feature style usually involves the statistical attributes of features, and existing studies [16] usually use the mean and standard deviation of features to represent feature style. Therefore, these two statistical attributes are adopted as feature styles in this study. Specifically, given the intermediate feature $f_{tir}^l \in \mathcal{R}^{B \times N \times D}$ from TIR student branch and $f_{rgb}^l \in \mathcal{R}^{B \times N \times D}$ from RGB student branch,

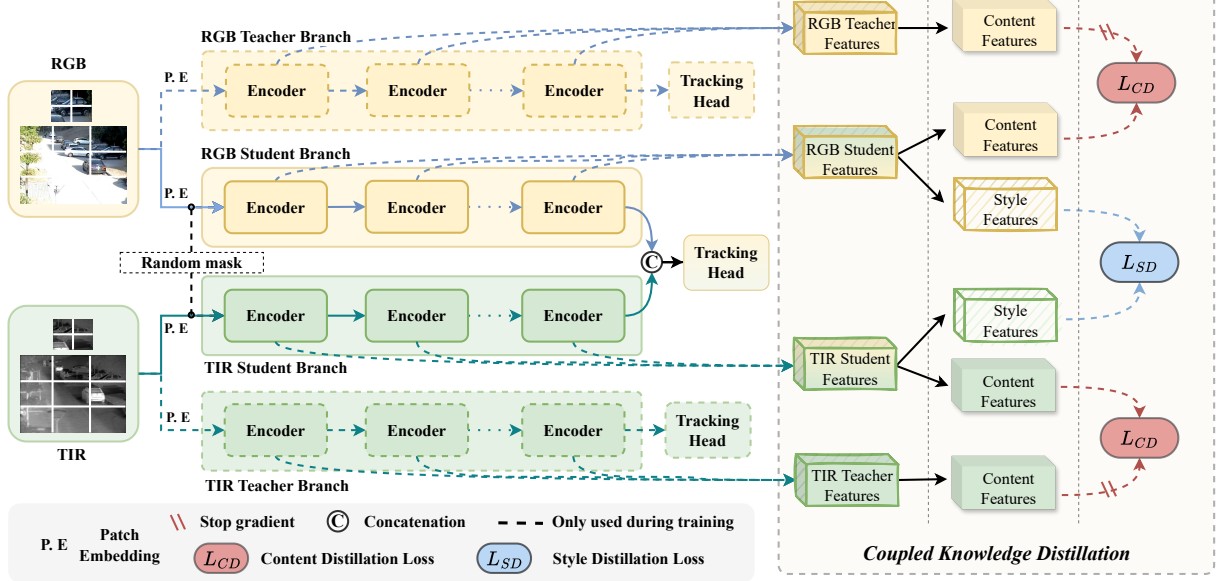

**Figure 3: Overall architecture of the proposed CKD. It mainly consists of a four-branch network, three tracking heads, and a coupled distillation framework. The four-branch network extracts visual features from the input video frames and performs style distillation and content distillation in the coupled distillation framework.**

where $l$ denotes the feature from the $l$-th layer, $B$, $N$, and $D$ represent the batch size, number of tokens, and channel dimension of tokens, respectively. For brevity, the $B$ dimension is omitted below. The process of calculating the mean and standard deviation vectors of the two student branch features is as follows:

$$\mu_{rgb}^{s(l)} = \frac{1}{N}\sum_{n=1}^{N} f_{rgb}^{l(n)}, \quad \sigma_{rgb}^{s(l)} = \sqrt{\frac{1}{N}\sum_{n=1}^{N}(f_{rgb}^{l(n)} - \mu_{rgb}^{s(l)})^2}, \quad (1)$$

$$\mu_{tir}^{s(l)} = \frac{1}{N}\sum_{n=1}^{N} f_{tir}^{l(n)}, \quad \sigma_{tir}^{s(l)} = \sqrt{\frac{1}{N}\sum_{n=1}^{N}(f_{tir}^{l(n)} - \mu_{tir}^{s(l)})^2}. \quad (2)$$

where $\mu_{rgb}^{s}$ and $\sigma_{rgb}^{s}$ represent the mean and standard deviation vectors of RGB student features, respectively, while $\mu_{tir}^{s}$ and $\sigma_{tir}^{s}$ indicate the corresponding vectors for TIR student features. Then, we can compute the style distillation loss, denoted as $\mathcal{L}_{SD}$, by quantifying the mean squared error between the style features across all layers of the two student branches.

$$\mathcal{L}_{SD} = \frac{1}{L}\sum_{l=1}^{L}((\mu_{tir}^{s(l)} - \mu_{rgb}^{t(l)})^2 + (\sigma_{tir}^{s(l)} - \sigma_{rgb}^{t(l)})^2). \quad (3)$$

By minimizing this style distillation loss during training, we can achieve style consistency across different modality features.

*3.2.2 Content Distillation.* Although RGB and TIR features of two student branches can eliminate modal gaps through style distillation, this process may harm the modal feature content representation. To alleviate this issue, we perform content distillation between teacher and student branches with the same modality input. To avoid imposing constraints on the style features in student branch,

we adopt the classical instance normalization operation to normalize teacher and student features, thus obtaining content features that are orthogonal to the style features. We can then calculate the similarity between the content features of the two groups of teachers and students for content distillation.

In particular, we take the TIR modality as an example and can describe above process as follows. Firstly, we perform feature instance normalization along the channel dimension as follows:

$$\hat{F}_{tir}^{l} = \frac{F_{tir_d}^{l} - \mu_{tir_d}^{t}}{\sigma_{tir_d}^{t}}, \hat{f}_{tir}^{l} = \frac{f_{tir_d}^{l} - \mu_{tir_d}^{s}}{\sigma_{tir_d}^{s}}. \quad (4)$$

where $F_{tir}^{l}$ denotes the intermediate feature from TIR teacher branch, $d$ represents the $d$-th channel dimension of each token. $\hat{F}_{tir}^{l}$ represents the $l$-th layer normalized feature (i.e., content feature) of TIR teacher branch, and $\hat{f}_{tir}^{l}$ denotes the $l$-th layer content feature of student branch. $\mu_{tir}^{t}$, $\sigma_{tir}^{t}$, $\mu_{tir}^{s}$, and $\sigma_{tir}^{s}$ correspond to the mean and standard deviation, respectively. Subsequently, we calculate the TIR feature content distillation loss, denoted as $\mathcal{L}_{CD}^{tir}$, by measuring the mean squared error (MSE) between the content features across all layers of teacher and student branches:

$$\mathcal{L}_{CD}^{tir} = \frac{1}{L}\sum_{l=1}^{L}(\hat{F}_{tir}^{l} - \hat{f}_{tir}^{l})^2. \quad (5)$$

Minimizing this loss during training ensures consistency between the content features of TIR student branch and those of TIR teacher branch. Similarly, this approach is applied in RGB student branching learning by incorporating a constraint from RGB teacher branch to maintain stability in content features. Therefore, the total content

distillation loss can be defined as follows:

$$\mathcal{L}_{CD} = \mathcal{L}_{CD}^{tir} + \mathcal{L}_{CD}^{rgb}. \tag{6}$$

where $\mathcal{L}_{CD}^{rgb}$ denotes the RGB feature content distillation loss.

In summary, coupled knowledge distillation is a combination of style distillation and content distillation that can break the modality gap in RGBT tracking. Thus, by incorporating CKD into our task, we can train two student branches that extract style-consistent features from different modalities while preserving their individual semantic content features.

## 3.3 Masked Modeling

Inspired by recent success and scalability of pretraining with masked reconstruction in different domains [4, 10], we design a novel masked modeling in the proposed framework to enhance modality content representation in challenging scenarios. Since there is natural feature content supervision between teacher and student branches of the same modality input in CKD, we can implement mask modeling without introducing any additional loss and design, simply by applying random mask to the input of student branches.

In particular, we randomly mask some of the input tokens of a student branch with $m \in \{0, 1\}^N$, where $N$ is the number of tokens. Hence the masked tokens are represented as $\{I_{rgb}^s | m_i = 1\}$ and $\{I_{tir}^s | m_i = 1\}$ while the remain tokens are denoted as $\{I_{rgb}^s | m_i = 0\}$ and $\{I_{tir}^s | m_i = 0\}$. Subsequently, we feed these tokens into the corresponding student branches, respectively. It is noteworthy that the input tokens for corresponding teacher branch remain unchanged. Consequently, we can utilize the token features from two teacher branches to guide the feature learning of the masked tokens in two student branches. In fact, the aforementioned process can be realized by minimizing the content distillation loss, which seeks feature content consistency between the masked student tokens and the teacher tokens. Therefore, the masked modeling can be seamlessly integrated into CKD. In addition, we set the mask ratio as 25% empirically.

## 3.4 Multi-modal Token Elimination

The effectiveness of using candidate elimination strategies to provide inference efficiency has been demonstrated in [48]. Specifically, existing RGB trackers determine which tokens to eliminate by leveraging the attention weights established by the tokens in target and search regions. However, this strategy ignores that the elimination results are unreliable when the input data are of poor quality. To address this issue, we propose a multi-modal candidate token elimination strategy, which aims to improve elimination quality in challenging scenarios by combining attention weights of two modalities through cooperative decision making. Given the query vector $q_r^{zi}$ from RGB template queries and the query vector $q_t^{zi}$ from TIR template queries, a scalar $h_i$ is assigned to each token in the search region, calculated as follows:

$$h = \max(\text{softmax}(q_r^{zi} k_r^x), \text{softmax}(q_t^{zi} k_t^x)) \tag{7}$$

where $k_r^x$ and $k_t^x$ represent the key vectors of tokens in the search region. We use $h$ to sort the tokens in the search region and keep the top-k tokens. Therefore, the proposed method not only enhances

the robustness of token elimination in challenging scenarios, but also improves the inference speed of the model.

## 3.5 Final Loss

In CKD, we can define the final loss function as a combination of content distillation, style distillation, and task losses, as follows:

$$\mathcal{L}_{all} = \mathcal{L}_{task} + \lambda_{cd} \times \mathcal{L}_{CD} + \lambda_{sd} \times \mathcal{L}_{SD}. \tag{8}$$

Here, $\mathcal{L}_{task}$ denotes the tracking task loss [48]. $\lambda_{cd}$ and $\lambda_{sd}$ are the loss coefficients corresponding to the two loss terms, respectively. In our study, we set the ratio of $\lambda_{sd}$ and $\lambda_{cd}$ to 2:1, respectively.

## 4 Experiments

### 4.1 Implementation Details

We choose OSTrack [48] as our foundational tracker, employing ViT [8] as its feature extractor. For parameter initialization, we utilize the original pre-trained model of OSTrack-base-256 [48]. For each sequence in a given training set, we collect the training samples and subject them to standard data augmentation operations, including rotation, translation, and gray-scale, aligning with the data processing scheme of the base tracker [48]. During training, the entire model utilizes AdamW to minimize the classification and regression loss functions. We use the LasHeR training set to train the entire tracking network in an end-to-end manner, which is used to evaluate GTOT [19], RGBT210 [24], RGBT234 [20], and LasHeR [23]. For the evaluation of VTUAV [31], we utilize the training set from VTUAV as the training data. In addition, we set the learning rate of the backbone network to 5e-6, and the tracking head to 5e-5. The CKD implementation is conducted on the PyTorch platform using two Nvidia A100 GPUs with 40G memory, and a global batch size of 40. The model fine-tuning takes 30 epochs that each epoch contains 60000 sample pairs. For VTUAV, we fine-tuned 5 epochs.

### 4.2 Evaluation Dataset and Protocol

*4.2.1 Dataset.* GTOT dataset is the earliest RGBT tracking dataset, consisting of 50 RGBT video sequences with a total of around 15,000 frames. However, the average sequence length of 150 frames in the dataset limits a comprehensive evaluation of model performance. *RGBT210* dataset expands the scope by including 210 pairs of RGBT video sequences, amounting to approximately 209.4K frames. *RGBT234* dataset is an upgrade from *RGBT210* consists of 234 highly aligned RGBT video pairs, totaling approximately 233.4K frames. Importantly, it provides more accurate bounding box annotations and annotations for 12 challenge attributes. *LasHeR* dataset is the largest RGBT tracking dataset, containing 1224 aligned video sequences with a total number of frames up to 1469.6K frames. It provides 245 test sequences and 979 training sequences, which can comprehensively evaluate tracking performance. *VTUAV* dataset collects RGBT data from UAV scenarios, expanding the application of RGBT tracking. Our experiments primarily focus on the short-term tracking subset of this dataset.

*4.2.2 Protocol.* In our study, we utilize precision rate (PR) and success rate (SR) as the main evaluation metrics for one-pass evaluation (OPE), which are commonly employed in current RGBT tracking

tasks. These metrics enable a quantitative analysis of tracking performance. PR evaluates the fraction of frames where the distance between the tracker's output position and the true bounding box value falls below a predetermined threshold. Note that, we set the threshold to 5 pixels for the GTOT dataset and 20 pixels for other datasets, thereby calculating a representative PR score. SR the percentage of successfully tracked frames whose Intersection over Union (IoU) is greater than a specified threshold, and then SR metric is calculated by varying the threshold and computing the Area Under the Curve (AUC) of the resultant curve. To eliminate these effects, the PR is normalized using the scale of the ground truth box to calculate the NPR. The normalized accuracy curve can be obtained by changing the normalization threshold, and the region under the normalized accuracy curve with the normalization threshold in the range of $[0, 0.5]$ is calculated as the representative NPR score.

## 4.3 Quantitative Comparison

We evaluate our algorithm on five popular RGBT tracking benchmarks and compare its performance with current state-of-the-art trackers. The effectiveness of our proposed method is demonstrated in Table 1, which provides a summary of the comparison results.

*4.3.1 Evaluation on GTOT dataset.* The comparison results on GTOT dataset are shown in Table 1. Compared to state-of-the-art trackers, our method exhibits superior performance in GTOT dataset, achieving gains over QAT [26] by 1.0% in PR. We further compare our method with CMPP [38], the best-performing tracker on the PR metric for this dataset. Although the PR of our method is 0.1% below that of CMPP [38], our CKD outperforms CMPP by 2.0% in SR. As for our low PR, we attribute this to the prevalence of small objects in the GTOT dataset, for which CMPP's feature pyramid strategy aggregates features across all layers to enhance the feature representation capabilities. Additionally, CMPP builds a historical information pool using external storage, improving the representation of the current frame. However, these strategies significantly affect the efficiency of CMPP, making our CKD approximately 90 times faster than CMPP.

*4.3.2 Evaluation on RGBT210 dataset.* As shown in Table 1, CKD exceeds almost state-of-the-art trackers in RGBT210 dataset. Compared to mfDiMP [31], which is the winner of VOT2019-RGBT, CKD achieves significant improvements in PR/SR with a gain of 9.8%/9.7%. Moreover, compared with TBSI [17], which is the second best-performing algorithms in terms of SR, CKD outperforms it by 2.9%/2.7% on the PR/SR metrics. We further compare CKD with QAT [26], which are the second best-performing algorithms in terms of PR, and CKD outperforms it by 1.6%/3.3% on PR/SR.

*4.3.3 Evaluation on RGBT234 dataset.* To further evaluate the effectiveness of CKD, we conduct a series of experiments on RGBT234 dataset, including overall and attribute-based comparison.

**Overall Comparison.** RGBT234 dataset is one of the most important datasets in the field of RGBT tracking, and also the dataset with the most evaluation results of existing algorithms. Therefore, we evaluate CKD against 24 state-of-the-art RGBT trackers on RGBT234 dataset. The evaluation results are presented in Table 1. CKD outperforms all state-of-the-art RGBT methods in PR/SR metrics, and obtains new SOTA results of 90.0%/67.4% in PR/SR.

Compared with QAT [26] and TATrack [39] algorithms, which are the second best-performing algorithms in terms of PR and SR, respectively, CKD outperforms them by 1.6% and 3.0% in both PR and SR metrics.

**Challenge-based Comparison.** We also present the results of CKD against the most advanced RGBT trackers available, including BAT [1], SDSTrack [14], TBSI [17], and ViPT [58], on different challenge subsets. The evaluation results are shown in 4, where the marks of each corner represent the attributes of the challenge subset and the highest and lowest performance under that attribute, respectively. These attributes include no occlusion (NO), partial occlusion (PO), heavy occlusion (HO), low illumination (LI), low resolution (LR), thermal crossover (TC), deformation (DEF), fast motion (FM), scale variation (SV), motion blur (MB), camera moving (CM) and background clutter (BC). The results show that our method performs best in all challenge subsets, and it proves that CKD has great potential in various complex tracking scenarios.

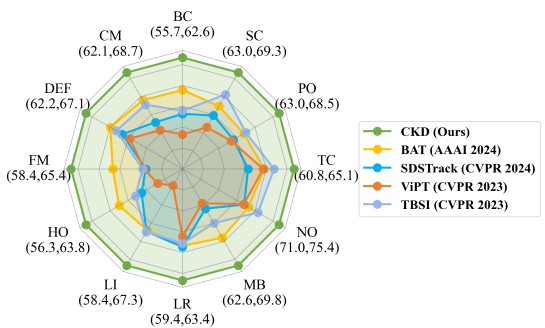

**Figure 4: Attribute-based evaluation on RGBT234 in terms of SR metric. CKD achieves the best performance on all attribute splits. Axes of each attribute have been normalized.**

*4.3.4 Evaluation on LasHeR dataset.* We compare 16 state-of-the-art RGBT trackers on LasHeR dataset, which is currently the largest and most challenging RGBT tracking dataset, and the evaluation results are shown in Table 1. CKD again outperforms all existing trackers by a clear margin. For instance, compared to BAT [1], the second best performing algorithm in this dataset, CKD exhibits a 3.0%/2.0% performance advantage on PR/SR metrics. The experiment further verifies the effectiveness of our approach in more complex scenarios.

*4.3.5 Evaluation on VTUAV dataset.* We evaluate the proposed method CKD on VTUAV dataset, a recently proposed drone perspective RGBT tracking dataset. From Table 1, it can be seen that CKD obtains 90.2%/77.8% on PR and SR metrics, which again confirm its effectiveness. Moreover, CKD surpasses the state-of-the-art MACFT [29] by 10.1% and 11.0% in PR and SR scores, respectively. The experiment further verifies the effectiveness of our approach in drone tracking scenarios.

## 4.4 Ablation Study

To verify the effectiveness of the proposed method, several ablation studies are performed on RGBT234 and LasHeR datasets.

**Table 1: PR/NPR and SR scores (%) for advanced trackers on five datasets.. The best and second are the result of the *red* and *blue*.**

| Methods | Pub. Info. | Backbone | GTOT PR↑ | GTOT SR↑ | RGBT210 PR↑ | RGBT210 SR↑ | RGBT234 PR↑ | RGBT234 SR↑ | LasHeR PR↑ | LasHeR NPR↑ | LasHeR SR↑ | VTUAV PR↑ | VTUAV SR↑ | FPS ↑ |
|---|---|---|---|---|---|---|---|---|---|---|---|---|---|---|
| MANet [22] | ICCVW 2019 | VGG-M | 89.4 | 72.4 | - | - | 77.7 | 53.9 | 45.5 | 38.3 | 32.6 | - | - | 1 |
| DAPNet [61] | ACM MM 2019 | VGG-M | 88.2 | 70.7 | - | - | 76.6 | 53.7 | 43.1 | 38.3 | 31.4 | - | - | 2 |
| mfDiMP [50] | ICCVW 2019 | ResNet-50 | 83.6 | 69.7 | 78.6 | 55.5 | - | - | 44.7 | 39.5 | 34.3 | 67.3 | 55.4 | 10.3 |
| CMPP [38] | CVPR 2020 | VGG-M | 92.6 | 73.8 | - | - | 82.3 | 57.5 | - | - | - | - | - | 1.3 |
| CAT [21] | ECCV 2020 | VGG-M | 88.9 | 71.7 | 79.2 | 53.3 | 80.4 | 56.1 | 45.0 | 39.5 | 31.4 | - | - | 20 |
| ADRNet [51] | IJCV 2021 | VGG-M | 90.4 | 73.9 | - | - | 80.7 | 57.0 | - | - | - | 62.2 | 46.6 | 25 |
| JMMAC [52] | TIP 2021 | VGG-M | 90.2 | 73.2 | - | - | 79.0 | 57.3 | - | - | - | - | - | 4 |
| MANet++ [27] | TIP 2021 | VGG-M | 88.2 | 70.7 | - | - | 80.0 | 55.4 | 46.7 | 40.4 | 31.4 | - | - | 25.4 |
| APFNet [45] | AAAI 2022 | VGG-M | 90.5 | 73.7 | - | - | 82.7 | 57.9 | 50.0 | 43.9 | 36.2 | - | - | 1.3 |
| DMCNet [28] | TNNLS 2022 | VGG-M | 90.9 | 73.3 | 79.7 | 55.5 | 83.9 | 59.3 | 49.0 | 43.1 | 35.5 | - | - | 2.3 |
| ProTrack [47] | ACM MM 2022 | ViT-B | - | - | - | - | 78.6 | 58.7 | 50.9 | - | 42.1 | - | - | 30 |
| MIRNet [13] | ICME 2022 | VGG-M | 90.9 | 74.4 | - | - | 81.6 | 58.9 | - | - | - | - | - | 30 |
| HMFT [31] | CVPR 2022 | ResNet-50 | 91.2 | 74.9 | 78.6 | 53.5 | 78.8 | 56.8 | - | - | - | 75.8 | 62.7 | 30.2 |
| MFG [41] | TMM 2022 | ResNet-18 | 88.9 | 70.7 | 74.9 | 46.7 | 75.8 | 51.5 | - | - | - | - | - | - |
| DFNet [30] | TITS 2022 | VGG-M | 88.1 | 71.9 | - | - | 77.2 | 51.3 | - | - | - | - | - | - |
| MACFT [29] | Sensors 2023 | ViT-B | - | - | - | - | 85.7 | 62.2 | 65.3 | - | 51.4 | 80.1 | 66.8 | 22 |
| CMD [53] | CVPR 2023 | ResNet-50 | 89.2 | 73.4 | - | - | 82.4 | 58.4 | 59.0 | 54.6 | 46.4 | - | - | 30 |
| ViPT [58] | CVPR 2023 | ViT-B | - | - | - | - | 83.5 | 61.7 | 65.1 | - | 52.5 | - | - | - |
| TBSI [17] | CVPR 2023 | ViT-B | - | - | 85.3 | 62.5 | 87.1 | 63.7 | 69.2 | 65.7 | 55.6 | - | - | 36.2 |
| QAT [26] | ACM MM 2023 | ResNet-50 | 91.5 | 75.5 | 86.8 | 61.9 | 88.4 | 64.4 | 64.2 | 59.6 | 50.1 | 80.1 | 66.7 | 22 |
| BAT [1] | AAAI 2024 | ViT-B | - | - | - | - | 86.8 | 64.1 | 70.2 | - | 56.3 | - | - | - |
| TATrack [39] | AAAI 2024 | ViT-B | - | - | 85.3 | 61.8 | 87.2 | 64.4 | 70.2 | - | 56.1 | - | - | 26.1 |
| OneTracker [12] | CVPR 2024 | ViT-B | - | - | - | - | 85.7 | 64.2 | 67.2 | - | 53.8 | - | - | - |
| Un-Track [44] | CVPR 2024 | ViT-B | - | - | - | - | 84.2 | 62.5 | 66.7 | - | 53.6 | - | - | - |
| SDSTrack [14] | CVPR 2024 | ViT-B | - | - | - | - | 84.8 | 62.5 | 66.5 | - | 53.1 | - | - | 20.9 |
| Ours | - | ViT-B | 93.2 | 77.2 | 88.4 | 65.2 | 90.0 | 67.4 | 73.2 | 69.3 | 58.1 | 90.2 | 77.8 | 96.4 |

**Table 2: Ablation study on the main components of CKD.**

| | Pretrained model | RGBT234 PR | RGBT234 SR | LasHeR PR | LasHeR SR |
|---|---|---|---|---|---|
| baseline | SOT | 86.4 | 64.5 | 67.8 | 54.0 |
| w/ SD | SOT | 86.4 | 65.0 | 68.9 | 54.5 |
| w/ SD CD | SOT | 87.4 | 65.5 | 71.6 | 56.9 |
| w/ SD CD MM | SOT | 88.6 | 66.1 | 72.3 | 57.4 |
| w/ SD CD MM | DropMAE | **90.4** | **67.8** | **73.1** | **58.0** |

*4.4.1 Component Analysis.* In Table 2, we conduct ablation studies on RGBT234 and LasHeR datasets to verify the effectiveness of different designed modules in CKD. Our baseline structure is the same as CKD, along with consistent training data and task losses, to fairly verify the effectiveness of the proposed components.

**w/ SD** denotes the baseline equipped with style distillation, which achieves a certain improvement. The experiment shows that aligning modality styles is effective, but there are limitations.

**w/ SD CD** indicates that adding content distillation to **w/ SD** results in significant performance improvements. The experiment shows that it is crucial to preserve the stability of modality content representation, as unconstrained style distillation could harm modality content representation, which could explain the limitations of **w/ SD**.

**w/ SD CD MM** represents that adding masked modeling to **w/ SD CD**. The experiment demonstrates the effectiveness of the masked modeling strategy.

*4.4.2 Impact of Pretrained model.* We also explore the DropMAE [43] pretrained model trained on the Kinetics700 dataset [2] as our pretrained model, which further achieves significant performance gains. Compared to the "SOT" pretrained model usually exploited by existing RGBT tracking methods [1, 12, 17, 44], DropMAE can bring superior performance to RGBT tracking. The experiment provides insights to further improve RGBT performance.

*4.4.3 Effectiveness of token elimination strategy.* To verify the effectiveness of the proposed multi-modal candidate token elimination strategy, we evaluate different token elimination methods in Table 3. Here, $CKD_{slow}$ represents the CKD method without a token elimination strategy, but it is still faster than existing RGBT trackers.

**w/ CE [48]** indicates that the two student branches individually apply the candidate token elimination strategy, following [48]. However, although CE brings an improvement in tracking efficiency, it also causes a significant performance drop.

**w/ MCE** indicates that the two student branches follow the proposed multi-modal candidate token elimination strategy for collaborative token elimination. It can be seen that MCE achieves a balance between tracking efficiency and accuracy.

*4.4.4 Hyper-parameter sensitivity analysis.* We analyze the parameter sensitivity as follows.

**Impact of loss weights.** We explore the influence of different loss weights between style and distillation losses on CKD performance in Figure 5. From Figure 5 it can be observed that the two kinds of losses in CKD are robust to the hyperparameters for these weights.

Andong Lu, Jiacong Zhao, Chenglong Li, Yun Xiao, & Bin Luo

**Table 3: Ablation study on the different elimination scheme.**

|  | RGBT234 | | LasHeR | | MACs(G) | FPS |
|---|---|---|---|---|---|---|
|  | PR | SR | PR | SR |  |  |
| CKD$_{slow}$ | **90.4** | **67.8** | 73.1 | 58.0 | 57.802 | 84.8 |
| w/ CE [48] | 88.7 | 66.5 | 73.0 | 58.0 | 42.735 | 96.4 |
| w/ MCE | 90.0 | 67.4 | **73.2** | **58.1** | **42.735** | **96.4** |

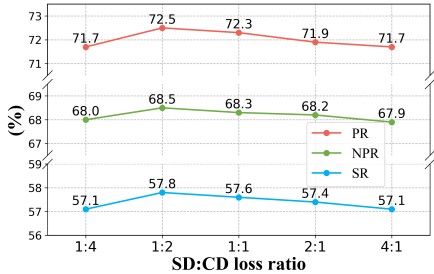

**Figure 5: Ablation study of loss weights on LasHeR dataset.**

**Table 4: Ablation study on different masked ratios.**

|  | RGBT234 | | LasHeR | | |
|---|---|---|---|---|---|
|  | PR | SR | PR | NPR | SR |
| CKD w/ mask 0% | 87.4 | 65.5 | 71.6 | 67.5 | 56.9 |
| **CKD w/ mask 25%** | **88.6** | **66.1** | **72.3** | **68.1** | **57.4** |
| CKD w/ mask 50% | 88.2 | 65.1 | 71.4 | 67.2 | 56.9 |
| CKD w/ mask 75% | 88.2 | 64.3 | 70.6 | 66.7 | 56.4 |

**Impact of masked ratios.** As shown in Table 4, we analyze the influence of different masked ratios on masked modeling strategy in CKD. It can be observed that the performance of CKD is always improved after the introduction of mask modeling, but the performance decreases slightly with the increase of mask modeling.

*4.4.5 Analysis of feature decoupling scheme.* In Table 5, we design several variants to verify the effectiveness of feature decoupling.

**baseline w/ IN** denotes the introduction of instance normalization in both student branches, which performs tracking with only content features. The experiment shows that modality style features certain discriminative information, which can lead to performance loss when directly dropped.

**baseline w/ FD** represents the introduction of non-decoupled feature distillation (FD) only between two student branches. The experiment suggests that the non-decoupled distillation scheme may harm the modality content representation.

**baseline w/ SD** is to perform distillation only in the style features between two student branches. The experiment further verifies that performing distillation for all modal features is unnecessary.

**baseline w/ CKD** is the coupled distillation scheme proposed in this paper. The experiment further demonstrating that the importance of feature decoupling scheme.

*4.4.6 Visual analysis.* In Figure 6, we visualize and compare the TIR content features extracted by the models trained with different distillation methods and not trained with distillation methods, and

**Table 5: Ablation study on the feature decoupling scheme.**

|  | RGBT234 | | LasHeR | | |
|---|---|---|---|---|---|
|  | PR | SR | PR | NPR | SR |
| baseline | 86.4 | 64.5 | 67.8 | 64.3 | 54.0 |
| baseline w/ IN | 85.6 | 63.7 | 67.1 | 63.2 | 53.4 |
| baseline w/ FD | 85.2 | 63.8 | 67.2 | 63.4 | 53.7 |
| baseline w/ SD | 86.4 | 65.0 | 68.9 | 64.3 | 54.5 |
| baseline w/ CKD | **87.4** | **65.5** | **71.6** | **67.5** | **56.9** |

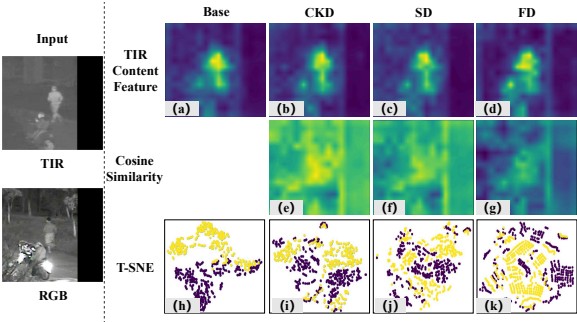

**Figure 6: Comparison of feature maps and T-SNE visualizations for different distillation methods. For T-SNE maps, they have the same scale of axes. The hotter color in the first row indicates more salient features, while in the second row the hotter color indicates more similar between the non-distilled (Base features) and distilled features, and vice versa. In the third row, the yellow and purple color indicate the features of RGB and TIR modalities respectively.**

show their similarity relationships. The experiment shows that the proposed CKD method achieves a good balance between modality gap elimination and modality content representation preservation.

## 5 Conclusion

In this work, we present a novel Coupled Knowledge Distillation (CKD) for RGBT tracking, which is the first effort to break the modality gap challenge in RGBT tracking. We first analyze the influence of modality style on modality gap, and then the proposed CKD can effectively enhance the consistency of modality style and avoid harming to modality content representation. Moreover, the proposed masked modeling strategy and a multi-modal candidate token elimination strategy effectively improve tracking performance and efficiency. Extensive experiments demonstrate the superiority of the proposed method. In the future, we will explore the benefits of the proposed CKD in other multi-modal visual tasks, such as RGBD/RGBE tracking [60, 63], image fusion [25, 55] and collaborative learning [46].

## Acknowledgments

This work was supported by the Natural Science Foundation of China (No. 62376004), and the Natural Science Foundation of Anhui Province (No. 2208085J18).

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
