# OpenReview forum: "Breaking Modality Gap in RGBT Tracking: Coupled Knowledge Distillation"
_acmmm.org/ACMMM/2024/Conference — MM2024 Poster_

### Official Review · Reviewer_12Mn · 2024-04-27
**Breaking Modality Gap in RGBT Tracking: Coupled Knowledge Distillation**

**Rating:** 3
**Confidence:** 4

**Review:**

This paper focuses on the modality gap between RGB and TIR modalities and tries to break this gap through the knowledge distillation.
After reading, my conclusion is displayed below:

Strengths:
1. The paper is well-structured and easy for understanding.
2. Various experiments are conducted.
3. Very good performance and efficiency.

Weaknesses:
1. The reasons for some of the designs are not clear: In introduction, the issue is the modality gap caused by the image style. It's reasonable to use techniques like style transfer, but the reason for using knowledge distillation is missed; For the content distillation, the authors claim that style distillation 'may' harm the feature embeddings.Although the performance is improved with content distillation, it still lacks further illustrations like visualisation of feature maps.
2.  Although the papes are limited, I believe that the qualitative analysis is not enough.
3. The most important problem is that it is not explained why we need to break the modality gap. In my opinion, we use the RGB and TIR data together because of the modality gap, which provides the complementary information.Furthermore, why breaking the modality gap can improve better performance.

Based on this, 'Marginally below acceptance threshold' is recommended since the most improtant part is not well explained. Also, the grade can be changed by explaining this well.

**Summary:**

This paper focuses on the modality gap between RGB and TIR modalities and tries to break this gap through the knowledge distillation.
It uses style and content transfers to achieve this goal. Besides, a token elimnination strategy is employed to discard the unhelpful tokens in challenging scenarios.

**Strengths:**

1. The paper is well-structured and easy for understanding.
2. Various experiments are conducted.
3. Very good performance and efficiency.
The whole manuscript will be much better if the following weaknesses are solved.

**Limitations:**

1. The reasons for some of the designs are not clear: In introduction, the issue is the modality gap caused by the image style. It's reasonable to use techniques like style transfer, but the reason for using knowledge distillation is missed; For the content distillation, the authors claim that style distillation 'may' harm the feature embeddings.Although the performance is improved with content distillation, it still lacks further illustrations like visualisation of feature maps.
2. Although the papes are limited, I believe that the qualitative analysis is not enough.
3. The most important problem is that it is not explained why we need to break the modality gap. In my opinion, we use the RGB and TIR data together because of the modality gap, which provides the complementary information.Furthermore, why breaking the modality gap can improve better performance. Based on this, 'Marginally below acceptance threshold' is recommended since the most improtant part is not well explained. Also, the grade can be changed by explaining this well.
4. Honestly, the similar thoughts of the modality gap can also be found in existing papers like MANet, MANet++ (modaliy-specific and modality-shared design), DFAT (data bias). So I don't think it's a new insight, and the authors also don't provide further analysis about this.

**Suitability:**

3

---

### Official Review · Reviewer_qREo · 2024-05-20

**Rating:** 4
**Confidence:** 3

**Summary:**

This paper focuses on RGB-Thermal tracking

**Strengths:**

The authors propose a novel coupled knowledge distillation framework CKD to handle the modality gap issue by eliminating
the style difference between RGB and thermal images for high performance RGBT tracking

**Limitations:**

Why can the style and content information be de-coupled ?
Can the proposed method be applied to other settings (such ad cross-domain image retrieval)?

**Suitability:**

2

---

### Official Review · Reviewer_VRnR · 2024-05-24

**Rating:** 4
**Confidence:** 3

**Summary:**

The paper addresses the challenge of the modality gap between RGB and TIR images in RGBT tracking. The proposed solution, a novel CKD framework, aims to harmonize the style features across modalities to enhance tracking performance. This approach leverages dual student networks and a style-content orthogonal feature decoupling scheme, maintaining the integrity of content representations while aligning style features. Extensive evaluations across multiple RGBT tracking datasets demonstrate the effectiveness and efficiency of CKD.

**Strengths:**

1. The paper is well-organized and articulates complex concepts clearly and concisely.
2. CKD uniquely addresses the modality gap by harmonizing the style features between RGB and TIR modalities while preserving their content integrity through a style-content orthogonal feature decoupling scheme.
3. The CKD framework is rigorously evaluated across five standard RGBT tracking datasets, demonstrating its superiority over existing state-of-the-art methods.

**Limitations:**

1. The paper extensively compares the proposed CKD framework against other state-of-the-art methods primarily based on knowledge distillation. A broader comparative analysis including non-distillation based approaches could provide a more comprehensive understanding of CKD's performance relative to the entire spectrum of RGBT tracking methodologies.
2. Although the paper reports impressive speeds of 96.4 FPS, the computational complexity and resource requirements are not thoroughly discussed. Given the dual-network architecture and the intricate distillation processes, the actual resource demand during training and deployment could be significant, which might limit its applicability in resource-constrained environments.
3. The evaluations focus on standard datasets, which may not fully represent challenging real-world conditions such as extreme weather, varying illumination, or occluded environments. The performance of the CKD framework under such diverse and challenging conditions remains less explored.
4. The paper could enhance its methodological rigor by providing a more detailed analysis of the sensitivity of the model's performance to various hyperparameters, such as the weights of the distillation and task losses.

**Suitability:**

3

---

### Official Review · Reviewer_4bui · 2024-05-24

**Rating:** 4
**Confidence:** 2

**Summary:**

This paper focuses on the modality gap between RGB and thermal infrared images. The authors propose a novel Coupled Knowledge Distillation framework called CKD, which aims to bridge the modality gap by pursuing common styles of different modalities. Since the distillation of style features might harm the content representations of the two modalities in the student networks, the authors use the original RGB and TIR networks as teachers. They distill their content knowledge into two student networks respectively through a style-content orthogonal feature decoupling scheme.

**Strengths:**

In the paper, Coupled Knowledge Distillation is used to reduce cross-modal differences. Style distillation makes the style features of the two modalities as consistent as possible, while content distillation ensures that the modality content representation remains stable. This approach achieves state-of-the-art (SOTA) performance in certain settings on some datasets.

**Limitations:**

1. The CKD framework comprises four branches: an RGB teacher branch, a TIR teacher branch, an RGB student branch, and a TIR student branch. The teacher branches typically use pre-trained large-scale models, which increases model complexity. Please analyze and compare the complexity of the models.

2. The intermediate features 𝑓_{𝑡𝑖𝑟}^l from the TIR student branch and 𝑓_{rgb}^l from the RGB student branch are essential. Could you explain the specific process of how these intermediate features are obtained within the framework?

3. In Formula (1), when calculating the standard deviation vectors of RGB student features 𝜎^{𝑠(l)}_{𝑟𝑔𝑏}, why is the  𝜇_{𝑟𝑔𝑏}^t used instead of the 𝜇_{𝑟𝑔𝑏}^s.

4. In Figure 3, how do the RGB student features and TIR student features become style features and content features without interfering with each other?

**Suitability:**

2

---

### Meta-Review · Area_Chair_x4dT · 2024-06-25

**Recommendation:** Accept (Poster)
**Confidence:** 5

**Metareview:**

The paper is fine from all reviewers, the responses well addressed the concerns from the reviewers.